image processing/pattern recognition/ biomathematics

networks, routing optimization, optimal transport

**Author for correspondence:**
Caterina De Bacco
e-mail: caterina.debacco@tuebingen.mpg.de

# Principled network extraction from images

## Diego Baptista and Caterina De Bacco

Max Planck Institute for Intelligent Systems, Cyber Valley, Tuebingen 72076, Germany

(iD) CDB, 0000-0002-8634-0211

Images of natural systems may represent patterns of network-like structure, which could reveal important information about the topological properties of the underlying subject. However, the image itself does not automatically provide a formal definition of a network in terms of sets of nodes and edges. Instead, this information should be suitably extracted from the raw image data. Motivated by this, we present a principled model to extract network topologies from images that is scalable and efficient. We map this goal into solving a routing optimization problem where the solution is a network that minimizes an energy function which can be interpreted in terms of an operational and infrastructural cost. Our method relies on recent results from optimal transport theory and is a principled alternative to standard image-processing techniques that are based on heuristics. We test our model on real images of the retinal vascular system, slime mould and river networks and compare with routines combining image-processing techniques. Results are tested in terms of a similarity measure related to the amount of information preserved in the extraction. We find that our model finds networks from retina vascular network images that are more similar to hand-labelled ones, while also giving high performance in extracting networks from images of rivers and slime mould for which there is no ground truth available. While there is no unique method that fits all the images the best, our approach performs consistently across datasets, its algorithmic implementation is efficient and can be fully automatized to be run on several datasets with little supervision.

## 1. Introduction

Extracting network topologies from images is a relevant problem in applications where the subject of the image has a network-like structure. For example, satellite images of rivers [1], neuronal networks [2,3], blood or vein networks [4–6], mitochondrial networks [7,8] or road networks [9–12]. Assuming this could be done automatically and quantitatively, practitioners would then be able to apply the mathematical study of networks to make quantitative analyses about the topological properties of

the system at study. In practice, given a raw image, for instance, a satellite image of a river embedded in a landscape, extracting a network requires identifying a set of nodes and a set of edges connecting them. While it might be relatively easy to perform this identification qualitatively, the challenge here is performing this extraction automatically, thus avoiding tedious manual extraction or specific domain knowledge and *ad hoc* tools. At the same time, this task should be scalable with system size and number of images as high-quality images are increasingly available and for larger systems. In addition, a qualitative intuition of the possible existence of a network behind an image is not enough to ensure that no degree of subjectivity is introduced owing to the observer's eye. For instance, two different observers might both perceive the presence of a network-like structure but distinguish two different sets of nodes, and thus two different networks behind the same image. Another challenge is indeed that of performing this extraction in a principled way so that the number of arbitrary choices in defining what the network is should be limited, if not completely absent.

Here, we present a method that addresses these issues by considering the framework of optimal transport. Specifically, inspired by a recently developed model to extract network topologies from solutions of routing optimization problems [13], we adapt this formalism to our specific and different setting. We start from a raw image as input and propose a model that outputs a network representing the topological structure contained in the image. The novelty of this method is that its theoretical underpinning relies on a principled optimization framework. In fact, a proper energy function is efficiently minimized using numerical methods, which results in an output network topology. This implies, in particular, that network extraction may not depend on the observer's eye, but rather can be automatically done by solving this optimization problem.

We study our model on real images from different fields, we focus in particular on ecology and biology and compare results with an algorithm that relies on standard image processing techniques, highlighting the main differences resulting from these two different approaches. In particular, our model allows for an automatic and principled performance of two tasks: filtering network redundancies and selection of edge weights. These are usually challenging tasks for image processing schemes, as they rely on some pre-defined parameter setting in input, while we obtain both directly in output with our model.

Many solutions for the problem of automatic network extraction from images have been proposed in computer vision, mainly relying on image-processing techniques [6,10–12,14–20], for instance segmentation [21–23], or junction-point processes [24]. The idea is to measure variation of intensity contrast in the image's pixels to highlight curve-like structures. Within this context, NEFI [25] is a flexible toolbox based on a combination of standard image-processing routines. A different approach, closer to the one considered in this work, is that of adopting some sort of optimization framework. For instance, Breuer & Nikoloski [26] considered an optimization problem where the goal is to minimize the total roughness of a path (a measure depending on the difference of weights in adjacent edges), in order to decompose a filamentous network into individual filaments. However, these usually rely on domain-specific optimization set-ups that cannot be easily transferred across domains. Another example is the ant-colony optimization scheme used to extract blood vessels from images of retinas [27]. They all suffer from the nondeterministic polynomial time-hardness of the problem, typical of routing optimization settings. Thus this type of approach relies on approximation techniques. Finally, another approach is that of biologically inspired mathematical models like the one of Tero *et al*. [28,29] that consider dynamical systems of equations that emulate network adaptability, like that observed for the *Physarum polycephalum* slime mould. Our model is also inspired by the feedback mechanism of a slime mould, which adapts the conductivity of the network edges to respond to differences in fluxes.

Our method relies on the formalism of optimal transport theory used to extract networks from solutions of routing optimization problems proposed in [13] and referred to as NextRout. This is made of three subsequent steps, but here we need only the last two, namely the pre-extraction and the filtering steps. While we refer to that work for all the mathematical details, here we describe the main principles behind this method and adapt it to images. The idea is inspired by the behaviour of the *P. polycephalum* slime mould. The body of this organism forms a network structure that flexibly adapts to the surrounding environment and the distribution of food sources displaced in it. This network grows with a feedback mechanism between two physical quantities: the conductivities of network edges and the flow passing through them, through dynamics that is described by a set of equations (sometimes referred to as 'adaptation equations'). In practice, the problem starts by assigning food sources in space and spreading the slime mould uniformly to cover the whole space. The dynamics regulates how the slime mould changes its body shape in time to reach the

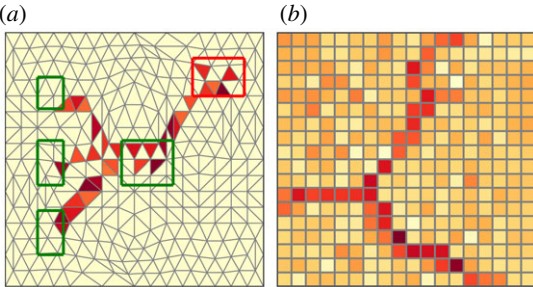

**Figure 1.** Analogy between optimizing trajectories and images. (*a*) The grid structure covering a continuous two-dimensional space and the optimal flux obtained by NextRout for a specific routing problem where sources and sinks are inside the green and red rectangles, respectively. Red tones of increasing darkness denote higher fluxes on the corresponding grid triangles. (*b*) The pixel grid and colours of a reference network-like image.

food in an efficient way. The stationary solution of this dynamical system is a set of conductivities and flows on edges that describe the optimal network topology covered by the mould. When the underlying space is continuous, like a squared patch in two dimensions, these solutions are functions defined on $(x, y)$ coordinates on this space. These are not immediately associated with a network meant as a set of nodes and a set of edges connecting them. However, Baptista *et al.* [13] propose principled rules to automatically extract network topologies from these solutions in continuous space. While the main focus of that work was to extract network topologies from this particular type of input (functions defined in a continuous domain, e.g. the space where food sources are located), they hinted at the possibility of adapting this formalism to discrete spaces, like images made of pixels. Here, we expand on this insight, and adapt this principled network extraction to inputs that are images. In particular, we propose an algorithm to effectively tackle two problems that are relevant for images and that were only briefly discussed for general applications in [13]: how to select source and sinks and how to obtain loopy structures.

## 2. Image2net: the method

The key idea is to treat the images as a particular discretization of a two-dimensional space by means of the pixels and treat the red-green-blue (RGB) colour values on them as conductivities. With this set-up, we can frame the problem as if there was an imaginary flow of colours. This starts by covering the whole discrete domain of the image uniformly and then flows through the pixels until it consolidates to a certain subset of them. The observed image corresponds to a network-like shape. Figure 1 illustrates the analogy between the solutions of NextRout in continuous space and an image of a network-like structure in discrete space.

As we mentioned before, this is not yet formally a network, as we do not have a rigorous definition of what constitutes a node and how nodes are connected. However, thanks to the analogy proposed here, we can use the rules introduced in [13] for the continuous case and adapt them to images. Specifically, we consider the pixels' centre of mass as nodes and draw edges between them depending on their pixels' locations and values, so that two nearby pixels are connected whenever the colour has a high enough intensity and their pixels are neighbours. We say that two pixels are nearby if they have a vertex or an edge in common (this corresponds to the pre-extraction scheme I, as explained in [13]). The result is a *pre-extracted network* [13] that we denote with $G^{pe}$. We denote with $V$ and $E$ the set of nodes and edges, respectively. The network is mathematically encoded by a signed incidence matrix which has entries $B_{ie} = \pm 1$ if the edge $e$ has node $i$ as start/endpoint, 0 otherwise. The sign is important to define the orientation of the flow passing through an edge.

This temporary network might contain redundancies like dangling nodes or redundant edges, see figure 2 for an example. Standard image processing techniques address this problem with pruning routines, e.g. by pruning away edges or branches shorter than a certain length. However, pruning has to be handled with great care, as small redundancies could be a major source of information or they could be completely irrelevant, depending on the network at hand. Usually, pruning is tuned by the user, thus creating potential for subjective bias in extracting the network. Instead, our model relies on a principled method for filtering such redundancies, which exploits a dynamics similar to that of the original problem but adapted to a discrete space like that of the network $G^{pe}$. However, to apply the

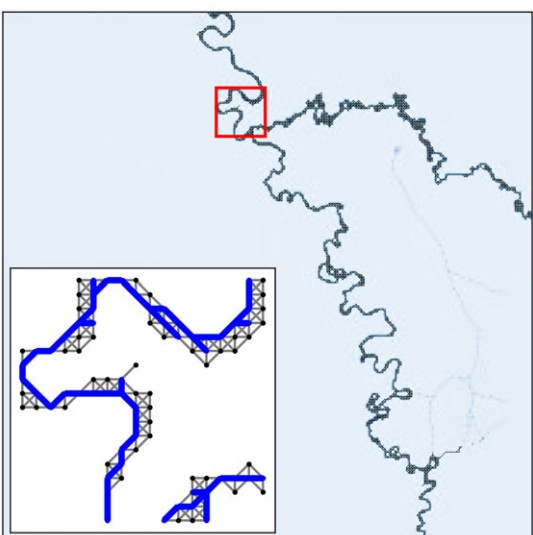

**Figure 2.** $G^{pe}$ taken from a river image. The subplot on the bottom left corner shows a section of $G^{pe}$ (in black), highlighted in red in the main plot, together with filtered graph $G^f$ (in blue).

filter, one must specify a set of terminals, sources and sinks, as input to the discrete dynamics. Continuing with our analogy, we need to locate the pixels where we imagine that colour mass is being injected and extracted. These are the sources and sinks that drive the dynamics to consolidate the flux of colours on the network-like structure observed in the input image.

## 2.1. Dynamics

Assume for a moment that we knew this set of terminal pixels and denote with $f_i$ the amount of colour mass that enters or exits the image in node $i$ (the centre of mass of pixel $i$). Note that to preserve the mass, we have $\sum_i f_i = 0$. Here we describe in more detail the dynamical rules that regulate how colours spread along the pixels in an optimal way. To describe the flow of colours, we consider the conductivity $\mu_e$ on an edge $e \in E$ and the potential $u_i$ on a node $i \in V$. The conductivities can be interpreted as proportional to the size of the diameter of an edge, and the potentials as pressures on nodes. Together, these two quantities determine the flow $F_e$ of colour passing through an edge $e = (v_1, v_2)$ in the network:

$$F_e = \frac{\mu_e}{\ell_e} (u_{v_1} - u_{v_2}) = \frac{\mu_e}{\ell_e} \left| \sum_{j \in V} B_{ej} u_j \right|. \tag{2.1}$$

The quantity $\ell_e$ denotes the Euclidean length of an edge using the centre of mass' coordinates. Although $\ell_e$ is either 1 or $\sqrt{2}$ for this particular choice of nodes, in the equations we keep $\ell_e$ for more general cases where the topology may not be as regular.

In turns, the flow influences the conductivities and potentials, through a feedback mechanism described by the following set of equations:

$$f_i = \sum_{e \in E} B_{ie} F_e, \tag{2.2}$$

$$\mu'_e(t) = \left[ \left| \frac{\mu_e(t)}{\ell_e} \sum_{j \in V} B_{ej} u_j(t) \right| \right]^\beta - \mu_e(t) \tag{2.3}$$

and

$$\mu_e(0) > 0, \tag{2.4}$$

where $| \cdot |$ is the absolute value and $\beta$ is a parameter that determines the optimization mechanism. Equation (2.2) is Kirchhoff's law; equation (2.3) is the discrete dynamics describing the feedback mechanism between conductivity and flow: when the flow of colours is high on an edge $e$, the conductivity increases, and vice versa when the flow is low the conductivity decreases; equation (2.4) is the initial condition. The stationary solution of this dynamical system can be mapped to the

solutions of an optimization problem where the cost function can be interpreted as a network transportation cost [13]:

$$\mathcal{L}_\beta(\mu(t)) = \frac{1}{2} \sum_e \mu_e(t) \left( \frac{1}{\ell_e} \sum_j B_{ej} u_j(\mu(t)) \right)^2 \ell_e + \frac{\beta}{2} \sum_e \frac{\mu_e(t)^{(2-\beta)/\beta}}{2-\beta} \ell_e, \tag{2.5}$$

where $\mu(t) = \{\mu_e(t)\}_e$, and the first term is the network operational cost and the second is the cost to build the network. The values of $\mu_e$ at convergence can be used not only to determine the set of edges in the extracted network but also its weights, which can be interpreted as proportional to the diameter of the edge on the image. This is one of the advantages of our model, as estimating the diameter of edges extracted from an image is an open problem when using image-processing techniques. We get this automatically with the optimal conductivities.

The dynamics works as a filter, i.e. removes redundancies, for $\beta \geq 1$. In this work, we fix $\beta = 1.5$ as it gives good performance consistently across the datasets studied here. The output result is a tree, i.e. it does not contain loops and is the optimal one in terms of minimizing the transportation cost of equation (2.5). In our experiments, we use the numerical solver proposed in [13] to extract the stationary solutions of the system of equations (2.2)–(2.4).

## 2.2. Selecting terminal pixels

Having introduced how the dynamics of the colours works, we now tackle the problem of selecting the terminal pixels where to inject or extract imaginary colour mass. This choice is crucial as it determines the final extracted networks. In the original problem of [13], this was not an issue because the set of terminals could be selected from that of the original problem in continuous space. In other words, this was an input of the problem. Here, we do not start from that input, but rather have access to only a raw image, without any notion of pre-specified terminals attached to it. In practice, we need to find the pixel nodes corresponding to the rectangles inside figure 1$a$. Here, we propose a method to make this selection effectively. Specifically, we select as a set of eligible terminals $\mathcal{T}(G_{\text{tree}}^{\text{pe}})$ all the leaves of the tree $G_{\text{tree}}^{\text{pe}}$ obtained from running our dynamics on $G^{\text{pe}}$ when we pass in input all pixel nodes in $G^{\text{pe}}$ as terminals, and selecting one of these at random as a source, all the rest as sinks. This choice is motivated by the fact that the tree resulting from the filtering is a good approximation of the pre-extracted $G^{\text{pe}}$, as it follows a principled optimization framework. The obtained leaves determine the coverage of this network, as they are usually located in distant parts of the network. Potentially, one could select terminal pixels 'manually', by using domain-knowledge to determine what pixels are the most important. However, this strategy is not scalable to a large number of images. Instead, our proposed procedure does not suffer from this problem as it can be automatically implemented, while also being flexible to receive 'hand-picked' terminals if available. Alternatively, a practitioner could make this selection based on some notion of network centrality, for instance selecting as terminal the most 'central' nodes. However, this again assumes having extra information to decide what definition of centrality is appropriate based on the application. We do not explore this here.

## 2.3. Obtaining loops

Running the dynamics of §2.1 outputs trees, while network-like structures in images might have loops. The question is thus how to recover networks that are not limited to trees. We tackle this problem by re-running the dynamics multiple times, each time selecting a particular node as source from the eligible ones (and sinks all the others). Specifically, we randomly select an individual source pixel $i \in \mathcal{T}(G_{\text{tree}}^{\text{pe}})$ and assign all the others $j \neq i \in \mathcal{T}(G_{\text{tree}}^{\text{pe}})$ as sinks. Applying the dynamics to $G^{\text{pe}}$ with this choice of one source and multiple sinks outputs a filtered network $G_r^f$, indexed by the iteration run $r$. By repeating for $N_{\text{runs}}$ this filtering step, each time selecting a different source from $\mathcal{T}(G_{\text{tree}}^{\text{pe}})$ (and all the remaining node pixels as sinks), results in a set of filtered networks $\{G_1^f, \ldots, G_{N_{\text{runs}}}^f\}$, all of them trees. We combine them by superposition, so that we obtain a final network $G(V, E, W)$ where the set of nodes and edges are the unions $V = \bigcup_{r=1}^{N_{\text{runs}}} V(G_r^f)$, and $E = \bigcup_{r=1}^{N_{\text{runs}}} E(G_r^f)$. The weights on the edges of the final network are given by the sum of the weights on each run:

$$w_{jk} = \sum_{r=1}^{N_{\text{runs}}} w_{jk}^r, \quad \forall j, k \in V, \tag{2.6}$$

where $w_{jk}^r$ is the weight of edge $(j, k)$ in network $G_r^f$ and corresponds to the optimal edge conductivities as obtained from the dynamics at convergence. We assume $w_{jk}^r = 0$, if $(j, k) \notin G_r^f$. The value of $N_{\text{runs}} \leq |\mathcal{T}(G_{\text{tree}}^{\text{pe}})|$ is a parameter that has to be tuned based on the input image. Note that a high value of $N_{\text{runs}}$ might not necessarily result in a network more similar to the one depicted in the input image. For instance, in the extreme scenario where the original network-like structure is a tree, then $N_{\text{runs}} = 1$. Empirically, we find that a value of $N_{\text{runs}} = 5$ gives good results in all the experiments reported here, see the electronic supplementary material for more details.

Combining these steps we obtain the whole algorithmic pipeline of our method, which we refer to as Image2net. It takes in input an image and it gives in output a network. We provide an algorithmic pseudo-code in algorithm 1 and an open-source implementation at https://github.com/diegoabt/Img2net.

Our algorithm can handle input images that display distinct objects, like parts of different rivers, resulting in separate network connected components. In fact, depending on how sources and sinks are selected, our algorithm can give in output naturally more than one connected component. Alternatively, one can apply the algorithm separately on each of the different connected components and then combine the results.

---

**Algorithm 1.** Image2net.

---

**Input:** Image $\mathcal{I}$, threshold $\delta$, $\beta \geq 1$

**Output:** $G(V, E, W)$ final network

1: **function** Image2net $\mathcal{I}, \delta, \beta \geq 1$

2:　　$G^{\text{pe}} \leftarrow$ NextRout pre-extraction$(\mathcal{I}, \delta)$

3:　　$G_{\text{tree}}^{\text{pe}} \leftarrow$ run Dynamics of §2.1 on $G^{\text{pe}}$　　▷ for $\beta$ and using in input as starting sources and sinks all the nodes in $G^{\text{pe}}$

4:　　$\mathcal{T}(G_{\text{tree}}^{\text{pe}}) \leftarrow \{v \in V(G_{\text{tree}}^{\text{pe}}) | d_v = 1\}$　　▷ $d_v$ is the degree of node $v$

5:　　**for** $r = 1, \ldots, N_{\text{runs}}$ **do**

6:　　　　Select $i \in \mathcal{T}(G_{\text{tree}}^{\text{pe}})$　　▷ uniformly at random

7:　　　　$G_r^f \leftarrow$ run Dynamics of §2.1 on $G^{\text{pe}}$　　▷ for $\beta$ and using as starting source $i$ and sinks $\{j \neq i \in \mathcal{T}(G_{\text{tree}}^{\text{pe}})\}$

8:　　**end for**

9:　　$G(V, E, W) \leftarrow$ Superimpose $\left\{ G_1^f, \ldots, G_{N_{\text{runs}}}^f \right\}$

10: **end function**

---

# 3. Experiments on images

We run our model on three datasets of images covering various types of network-like topologies observed in biology and ecology. The images represent: (i) the slime mould *Physarum polycephalum* (*Physarum polycephalum*) [30], which is also the inspiration of our dynamics; (ii) the retinal vascular system (*retina*) [31]; (iii) river networks (*rivers*) obtained by extracting images from [32]. The number of images taken from the *Physarum polycephalum*, *retina* and *rivers* sources is 25, 20 and 10, respectively, see table 1. Pre-processing was applied, see the electronic supplementary material for details.

For model comparison, we consider NEFI, a routine that combines various image processing techniques, and a variant of our routine based on a combination of minimum spanning tree and Steiner tree optimization (Image2net-MST). The idea behind this last routine is to run our procedure but replacing the optimization steps based on the dynamics of §2.1 with standard routing optimization algorithms, namely a combination of minimum spanning and Steiner trees [34]. The goal is to see how the underlying optimization set-up impacts the final network topology. In fact, while the core idea of treating the problem of network extraction from images within the framework of routing optimization is the same for Image2net and Image2net-MST, the details of their corresponding optimization differ. Specifically, for Image2net-MST, we first run a standard MST optimization algorithm to extract $G_{\text{tree}}^{\text{MST}}$ from $G^{\text{pe}}$ (this is the same input given to Image2net). From the corresponding set of leaves $\mathcal{T}(G_{\text{tree}}^{\text{MST}})$, this time one should extract a subset of terminals $T \subseteq \mathcal{T}(G_{\text{tree}}^{\text{MST}})$

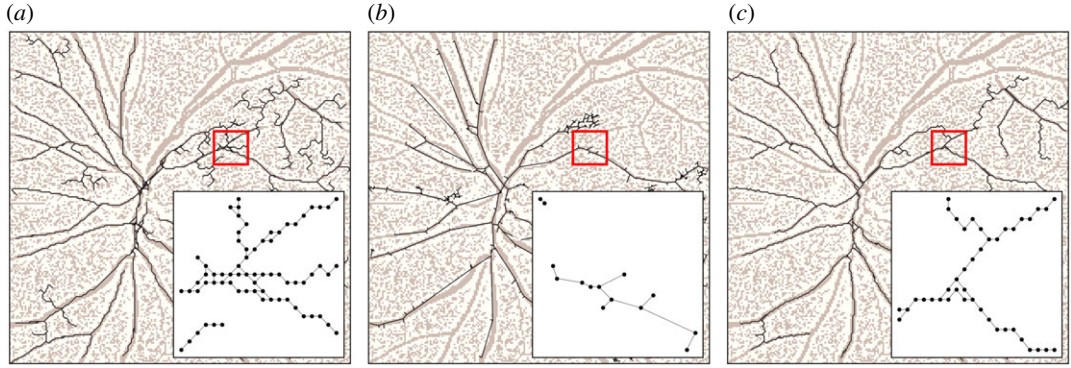

**Figure 3.** Example of network extraction. (*a*) Image2net; (*b*) NEFI; (*c*) Image2net-MST. The extracted network is coloured in black, the original image is in light brown underneath. The inset is a zoom inside the image section highlighted in red.

**Table 1.** Datasets description. (NI is the number of images used; AW is the average width of the images in the dataset; MinW and MaxW denote, respectively, the minimum and maximum width of the images in the dataset.)

| dataset | description | NI | AW | (MinW,MaxW) | ref. |
|---|---|---|---|---|---|
| retina | retinal blood vessels | 20 | 1791 | (998,2302) | [31,33] |
| Physarum polycephalum | slime mould | 25 | 400 | (400,400) | [30] |
| rivers | riverbed | 10 | 924 | (718,958) | [32] |

of predefined size (no distinction between source and sinks is necessary to solve a minimum Steiner tree problem). From $G^{pe}$ and $T$, extract a Steiner tree $G_r^{St}$, repeat this $N_{runs}$ times and obtain the set $\{G_1^{St}, \ldots, G_{N_{runs}}^{St}\}$. Finally, superimpose them as done for our method to obtain $G_{MST}$, see the electronic supplementary material for more details. Note that Steiner tree optimization has a complexity that scales with the number of terminals, a problem not present in our dynamics. As a result, running Image2net-MST is noticeably computationally more expensive than Image2net.

Finally, edge weights were assigned with rules specific to each method, as there is no common definition that applies to all of them. In fact, the ability to extract edge weights is rare among image processing techniques, and usually relies on image preprocessing and segmentation of the input image. Instead, Image2net extracts edge weights in a principled way based on the results at optimality in terms of conductivity, hence it has a nice direct interpretation as the diameter of the edges in the image. For Image2net, we use the rule effective reweighing (ER) on the resulting conductivities, see [13]; for Image2net-MST, we use the weights given in input to solve the Steiner tree problem, i.e. the weights given by ER rule in $G^{pe}$; for NEFI, we use as weight the *width*, this is an output of the algorithm; for the original image, we assign the RGB values of the pixels mapped into an integer number increasing with the colour intensity (see the electronic supplementary material for details). All of these definitions of weight agree on the higher the weight the thicker the edge is, and thus the conductivity. Figure 3 illustrates an example of the networks extracted using the various algorithms for an image in *retina*.

## 3.1. Performance metrics

We measure performance in terms of the ability of an algorithm to recover the network-like subject depicted on the underlying image. We consider a measure of similarity adapted from the quality measure defined in [13]. This relies on partitioning the image in a grid of $P$ non-intersecting subsets $C_\alpha$ inside the pixels' domain and then compare the edges of $G$ within $C_\alpha$ assigned by the algorithm and those observed in the original image $I$ (RGB values):

$$\hat{w}_b(G, I) = \frac{1}{P} \left[ \sum_{\alpha=1}^{P} \left( \left| \sum_{e \in E} \mathbb{1}_\alpha(e) - \sum_{i \in I} \mathbb{1}_\alpha(\delta, i) \right| \right)^2 \right]^{1/2}, \tag{3.1}$$

where $\mathbb{1}_\alpha(\delta, i) = 1, 0$ for $i \in I$, if the pixel $i$ is in $C_\alpha$ or not, respectively; $\delta$ is a threshold used to decide whether that pixel contributes to the network-like image. In words, if the pixel colour intensity is high

enough, then we label it as an edge. For $\delta$, we use the same value as used in input to Image2net. This is a coarse-grained measure of similarity that tells how many edges in the extracted graph correspond to high-intensity pairs of pixels. In order to account for edge weights and pixel intensities, we also consider a weighted version of this:

$$\hat{w}(G, I) = \frac{1}{P} \left[ \sum_{\alpha=1}^{P} \left( \left| \sum_{e \in E} \mathbb{1}_\alpha(e) w_e - \sum_{i \in I} \mathbb{1}_\alpha(i) p_i \right| \right)^2 \right]^{1/2}, \tag{3.2}$$

where $p_i$ is the intensity of the pixel $i$, and $\mathbb{1}_\alpha(i)$ is 1 if $i \in C_\alpha$, and 0 otherwise. Note that in this case $\delta$ is not needed because pixels with low intensity are penalized by lower weight in their contributions to $\hat{w}(G, I)$. In both cases, small values of these measures mean higher similarity values between the extracted network and the underlying network-like structure in the image.

While ground-truth for this network-like structure is normally absent, the *retina* dataset contains ground-truth networks which were hand-labelled by individuals [31]. In this case, we calculate the binary similarity using the hand-labelled images instead of the one given in input. There are two sets of labelled images, each corresponding to a different person doing this manual identification. While similar, the resulting two sets of networks are different. In the absence of ground truth, we compare against the input image.

## 3.2. Implementation details

We apply image pre-processing to the input image to improve image quality and distinguish the main subject from the background, see the electronic supplementary material for details. We rescale NEFI's pixels' location to have them in the same scale as that of the other methods, i.e. the set $[0, 1] \times [0, 1]$ (for simplicity, we consider only square-shaped images). All the edge lengths $\ell_e$ have been assigned using the Euclidean distance between the corresponding endpoints. For NEFI, we used the two best performing pipelines of image-processing techniques *polycephalum_high* (`NEFI-high`) and *crack_patterns* (`NEFI-crack`) among the available predefined pipelines. In the figures, we show the best results only, these vary based on the image given in input.

# 4. Results

## 4.1. Retinal vessel image validation

We use the similarity measure defined in the previous section to compare every graph-based approximation of the image with the provided hand-labelled ones, assuming these last ones to be the ground truth $I_{gt}$. We compute $\hat{w}_b(G, I_{gt})$ for each retinal image and the corresponding extracted network, to measure how close a particular network is from the human-labelled one. Figure 4 shows that Image2net consistently outperforms NEFI over all images and the two hand-labelled datasets. Image2net and Image2net-MST perform similarly according to the binary similarity. However, if we account for weight, we obtain the Image2net outperforms Image2net-MST in the majority of the images. Note that Image2net-MST does not assign new weights while selecting the edges, as in a Steiner tree problem, instead, it uses the weights of the input network, in this case, $G^{pe}$. Instead, Image2net selects edges and weights at the same time, within the same optimization set-up. The fact that the weighted similarity gives better results, signals that the values of the optimal conductivities (the weights assigned to Image2net extracted networks) have a meaningful interpretation, as they better match the pixel's intensities than the weights given by the other algorithms.

## 4.2. *Physarum polycephalum* and *rivers* networks

We measure the performance in the two datasets where there is no ground truth, which is often the case in real images. We find that Image2net recovers better the *rivers* networks, for both performance metrics as we show in figure 5. In fact, our model is able to capture the detailed geometry of the network when there are curves, while NEFI has limitations in that edges with curves or kinks are contracted to straight lines. This is one of the main advantages of our model based on an underlying optimization framework, the geometry of the network is automatically selected based on optimality, rather than a predefined setting manually tuned. As a result, Image2net is flexible in detecting different network geometries, as

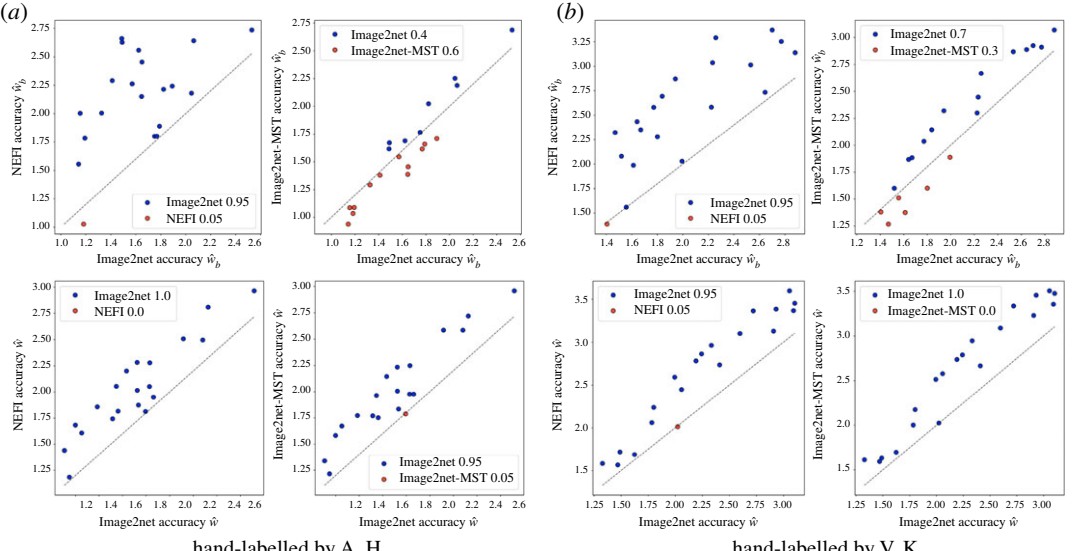

**Figure 4.** Recovering hand-labelled networks. Performance in terms of similarity $\hat{w}_b(G, I)$ and $\hat{w}(G, I)$ on hand-labelled *retina* networks, (*a*) and (*b*) are networks labelled by two different people. First row shows $\hat{w}_b$ values; second row shows $\hat{w}$ values. Smaller values mean higher similarity and thus better performance. Hence, points above the grey line (blue) means Image2net performs better, whereas points below (red) means worse performance. (*a*) Hand-labelled by A. Hoover (*b*) Hand-labelled by V. Kouznetsova.

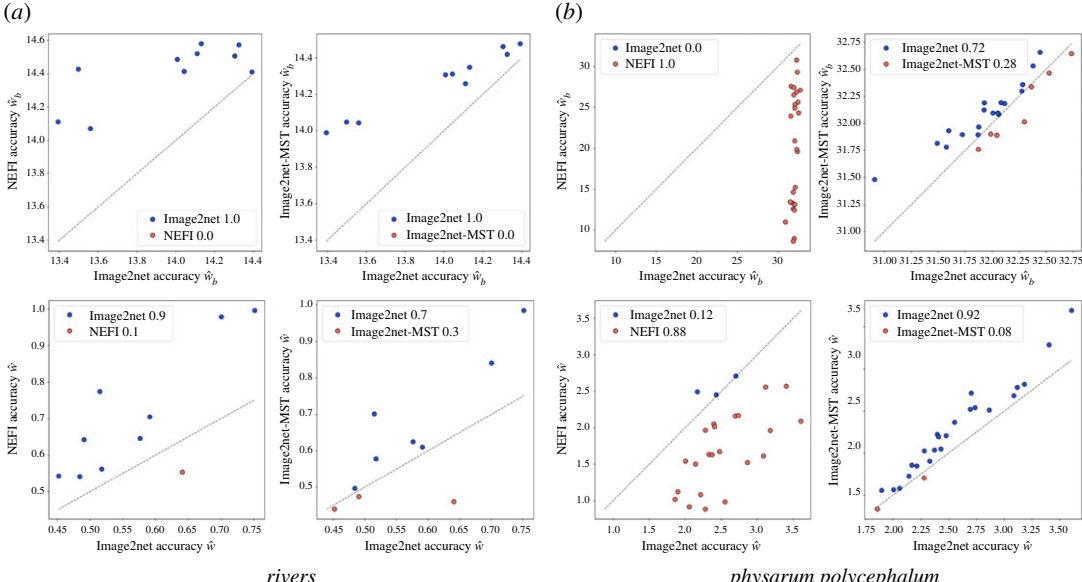

**Figure 5.** Recovering *river* and *Physarum polycephalum* networks. Performance in terms of similarity $\hat{w}_b(G, I)$ and $\hat{w}(G, I)$ on *rivers* (2 leftmost columns) and *Physarum polycephalum* (two rightmost columns) networks. Smaller values mean higher similarity and thus better performance. Hence, points above the grey line (blue) means Image2net performs better, whereas points below (red) means worse performance. (*a*) *rivers* and (*b*) *Physarum polycephalum*.

can be seen in figure 6 (top). The situation for *Physarum polycephalum* is more nuanced as Image2net is better than Image2net-MST, in particular when considering the weights, but NEFI outperforms all the others. However, this is true if we use the `NEFI-high` routine, which is the one built on purpose to detect *Physarum polycephalum* networks, it is not surprising that this has stronger results on these datasets. In figure 6 (bottom), we note how these networks contain many small details that are better captured by NEFI. Indeed, using other NEFI routines, performance aligns more to Image2net and Image2net-MST. This also shows that if a practitioner aims at extracting networks from a particular image, all the approaches allow for few degrees of freedom to be tuned in order to increase performance. NEFI allows the specification of individual routines to design a custom pipeline,

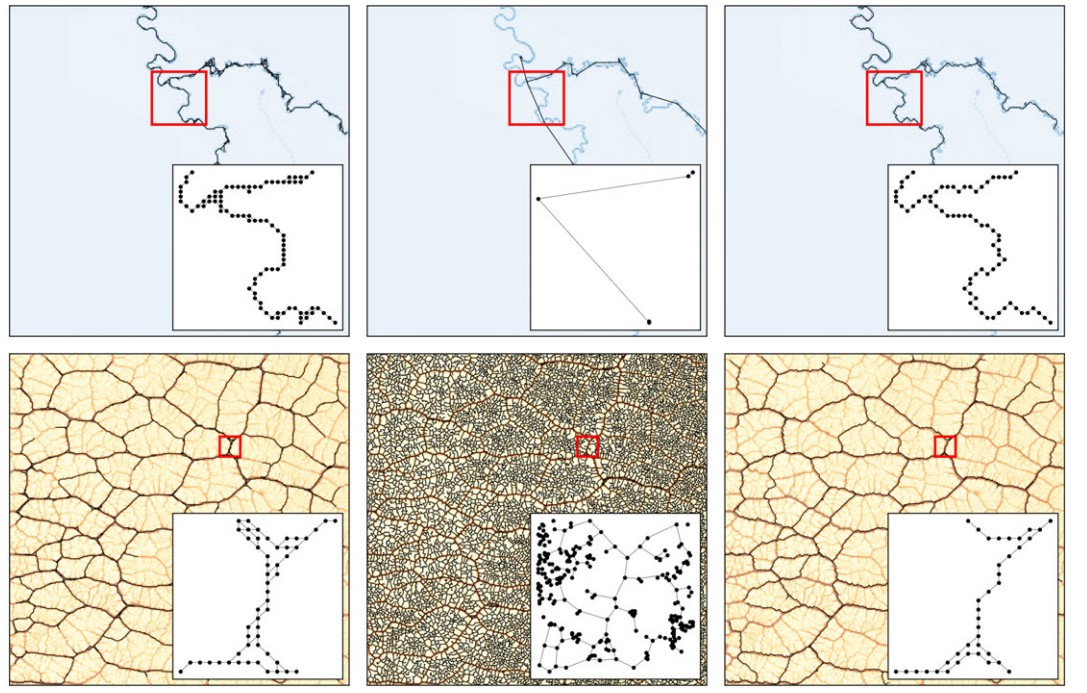

**Figure 6.** Results on *rivers* and *Physarum polycephalum* networks. We show the networks extracted on *rivers* (top) and *Physarum polycephalum* (bottom) using Image2net (left), NEFI (centre) and Image2net-MST (right). Inset is the zoom over the area under the red surface. The input image is depicted underneath the networks.

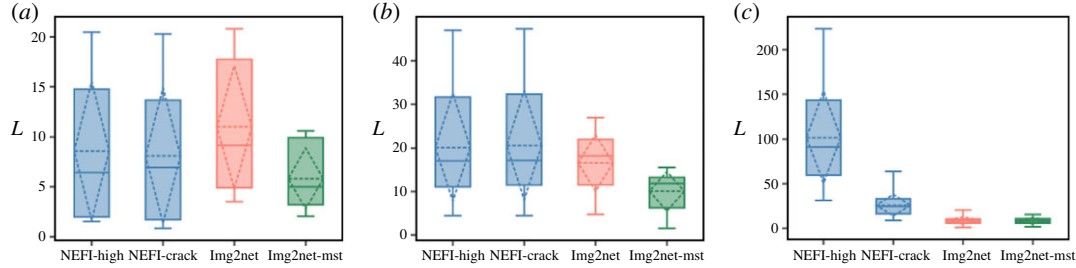

**Figure 7.** Total network length. Boxes show the distribution over the images inside each dataset of the total network length $L = \sum_e \ell_e$ calculated on the networks extracted by each method. Solid lines are the median, dashed lines are the average. (a) *rivers*, (b) *retina* and (c) *Physarum polycephalum*.

Image2net and Image2net-MST have various parameters that could be tuned, the most important being $\delta$ and $\beta$. For instance, decreasing $\delta$ will allow for more fine details on *Physarum polycephalum* networks, see the electronic supplementary material. However, tuning each routine on each input image goes beyond the scope of this work, as we aim at describing how different approaches perform on a corpus of images, potentially quite different, and thus automatize network extraction in a scalable way.

## 5. Qualitative results

Beyond validating the model on recovering network structure that resembles well what is pictured in an image, we illustrate the differences in topological properties of the extracted networks. This also showcases possible applications for our model, where a practitioner extracts a network and can then perform further analysis on it, for instance using the detected network properties.

We calculated the total network length as $L = \sum_e \ell_e$ where $\ell_e$ is the Euclidean distance between the nodes defining the edge $e$, see figure 7. We find that Image2net extracts on average longer *rivers* networks, and similar to NEFI for the *retina*, but with lower variance in this case. Instead, NEFI extracts much longer *Physarum polycephalum* networks, mainly owing to many small minor paths permeating the whole image (this was signalled above by wider result difference in terms of similarity). Instead,

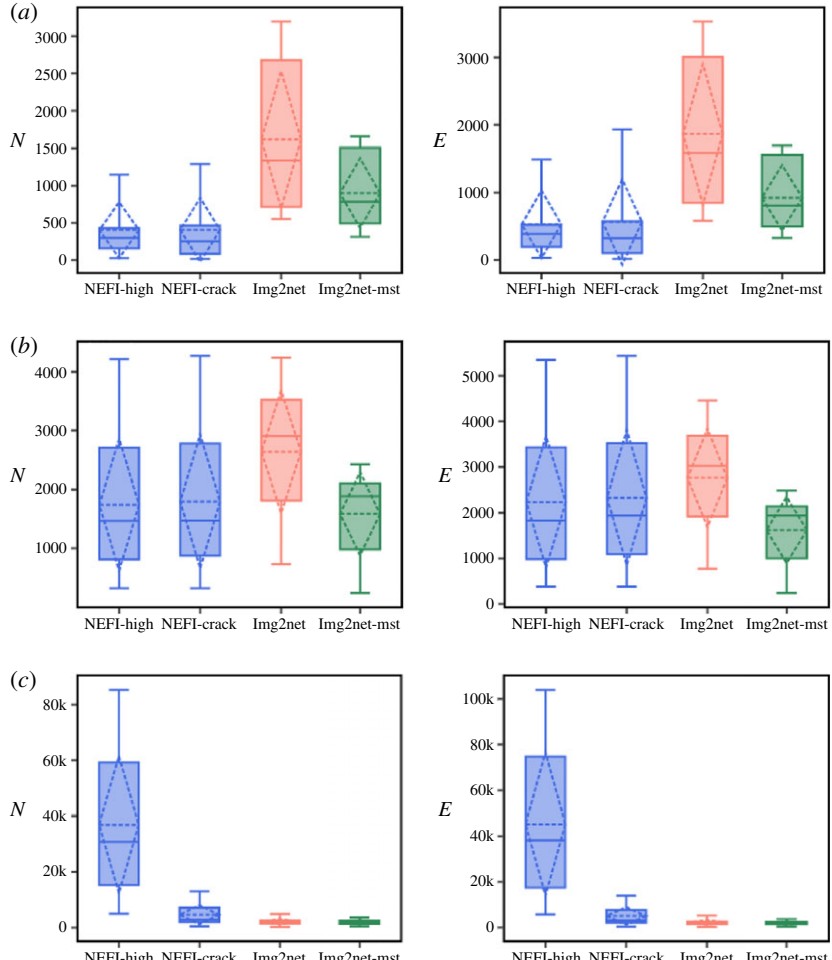

**Figure 8.** Number of nodes and edges. Boxes show the distribution over the images inside each dataset of $N$, the total number of nodes (*a*) and $E$, the total number of edges (*b*), calculated on the networks extracted by each method. Solid lines are the median, dashed lines are the average. (*a*) *rivers*, (*b*) *retina* and (*c*) *Physarum polycephalum*.

Image2net-MST finds smaller values in all the datasets. This highlights one important difference owing to the underlying optimization set-up that distinguished these two approaches.

While a longer network total length might be owing to a higher number of edges, this is not always the case. This can be seen from results on *rivers* in figure 8, where we plot the distribution of the number of nodes and edges, other important topological properties. In fact, for these images, NEFI finds much smaller network sizes than Image2net, while the distribution of $L$ in figure 7 is similar for the two routines. This is again owing to NEFI representing curved parts of the network with fewer but longer straight edges, see figure 6 for an example. In these river networks, Image2net has a higher resolution, where NEFI fails to find enough details. The opposite extreme is seen for the *Physarum polycephalum* images where NEFI has many more nodes and edges when using the routine `NEFI-high`, and also much higher $L$ as we saw before. For the *retina* vessel networks, Image2net extracts on average networks with higher number of nodes and edges than the other two methods, while $L$ is similar to NEFI, hence both Image2net and Image2net-MST have on average shorter edges than NEFI, with the difference that Image2net extract networks with bigger sizes.

## 6. Conclusion

We propose Image2net, a model for extracting networks from images. It takes as input an image and returns a network structure as a set of nodes, edges and the corresponding weights. Standard approaches for addressing these problems rely on image processing techniques. Instead, our model is based on a principled formalism adapted from recent results of optimal transport theory. We build an analogy with fluid dynamics by treating colours on pixels as fluids flowing through an image and

considering a set of dynamical equations for their conductivities and flows. At convergence, these correspond to stationary solutions of a cost function that has a nice interpretation in terms of a transportation cost.

The advantage of our approach with respect to more conventional methods is that our model naturally incorporates a principle definition of edge weights as the optimal conductivities and that can be interpreted as proportional to the diameters of network edges. In addition, it allows for a principled and automatic filtering of possible redundancies by means of solving a routing optimization problem, instead of using pre-defined pruning routines.

We test our model on various datasets, and calculate performance measures in terms of recovering the network-like shape in the input image. Image2net performs well compared to other network extraction tools and yields networks that closely approximate the networks depicted in the images. In particular, it is flexible in finding various network shapes, as it can find curved geometries as those observed in river networks.

We expect our method to be appropriate for images that represent phenomena with well-defined and physically meaningful flows or fluxes, as in the datasets considered here. In particular, for underlying conserved incompressible flows as water or blood flows. However, not all networks shown in images represent flow, for instance, fracture networks, foams, or grain boundaries in materials. While our method is still applicable to these scenarios, as it is agnostic to what image is given in input, the resulting networks may not be meaningful in these cases. In addition, there may be also flow-based systems displaying many loops or dynamical behaviours changing frequently in time, e.g. tidal marshes, which may not be captured well by our model. A possible solution for obtaining many loops could be to adapt Image2net to a multi-commodity optimal transport approach as in [35], which can naturally lead to loopy structures. We thus encourage practitioners to consciously select the images given in input to our algorithm based on the expected behaviour of the underlying physical phenomena being displayed.

In addition to being efficient, automated network extraction also has the advantage of yielding reproducible results and reducing human biases. Indeed, given an input image, Image2net will always yield the same networks, whereas manual extraction depends on the perception of the individual performing the measurement. Our model also enables practitioners to measure network-related quantities like centrality measures, branching points or curvature and angles. More importantly, given the computational efficiency of the underlying solver, it also works for large networks where manually measuring metrics across the whole network is not feasible.

In this work, we mostly show example applications from biology and ecology, but the usage of our model is not limited to this kind of networks. It can be used in a broad array of datasets to detect and measure network-like shapes, in particular, those displaying systems with incompressible flows. We foresee that our model will be useful for practitioners willing to perform automatic and scalable network analysis of large datasets of images.

Data accessibility. All data needed to evaluate the conclusions in the paper are present in the paper and/or the electronic supplementary material [36]. An open-source implementation of the code is available online at https://github.com/diegoabt/Img2net.

Authors' contributions. D.B. and C.D.B. derived the model, analysed results and wrote the manuscript. D.B. conducted the experiments.

Competing interests. The authors declare that they have no competing interests.

Funding. No funding has been received for the article.

Acknowledgements. The authors thank the International Max Planck Research School for Intelligent Systems (IMPRS-IS) for supporting Diego Baptista.

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
