## [Peer Review File · Royal Society Open Science]

Review History

RSOS-210025.R0 (Original submission)

Review form: Reviewer 1 (Sushil Kumar)

Is the manuscript scientifically sound in its present form?

No

Are the interpretations and conclusions justified by the results?

Yes

Is the language acceptable?

Yes

Do you have any ethical concerns with this paper?

No

Have you any concerns about statistical analyses in this paper?

No

Recommendation?

Major revision is needed (please make suggestions in comments)

Comments to the Author(s)

Author(s) proposed Image2net model to extract networks from images. The topic is interesting but my comments on this article as follows:

1. add some more references related to network building from images.
2. Refine the introduction section in the standard format.
3. You may consider some statistical metrics for evaluating the performance of the image2net model.
4. How network gives the correct information? pls. give theoretical reason about getting the correct network. There are various deep learning technique like autoencoders for prediction the unlabeled images.
5. Need more discussion on results.
6. Also mention the limitation of the proposed model.

Review form: Reviewer 2

Is the manuscript scientifically sound in its present form?

Yes

Are the interpretations and conclusions justified by the results?

Yes

Is the language acceptable?

Yes

Do you have any ethical concerns with this paper?

No

Have you any concerns about statistical analyses in this paper?

No

Recommendation?

Accept with minor revision (please list in comments)

Comments to the Author(s)

The article of Baptist and De Bacco describes a method of extracting a network from an image, where the network represents the connections between the key features in the image. This is a challenging problem and one with wide application. The methods presented are, in my experience, relatively innovative, and I would recommend publication in Open Science after addressing some concerns about the presentation of the method and results in the manuscript.

Most significantly, there are some quite strong assumptions in the way the image processing proceeds, which mean that it will be most appropriate to a particular class of phenomena with well-defined and physically meaningful flows or fluxes in it. Indeed, the three sample data sets include this (shuttle-flow in slime mold, blood flow in the retina, water flow in rivers; all nice conserved incompressible flows, with all the constraints that that implies). It would help potential users of this method to be explicit about this, for example in the abstract, intro and conclusions. While the methods here may be useful for other systems, the way the processing works really relies on mimicking the underlying transport, yet this is not stated clearly. Not all networks represent flows – fracture networks for example are well studied, or foams, or grain

boundaries in materials. Even flow-based systems may also not be as well-defined as those studied here, for example tidal marshes, where the flow reverses several times a day (and which are more loopy than normal river/stream networks).

Another aspect of the processing that deserves comment, especially given some of the data sets used, is how it handles images with distinct (i.e. not topologically connected) objects in it. For example, an image containing parts of two different rivers. Unless I have missed something important, and the images provided in the SI don't seem to suggest that I have, the entire network found must belong to a single connected cluster of points. If this is indeed correct, it really needs to be discussed, as it will affect the applications that this code can be applied to.

For the method, the way it is presented is often unclear, or hard to follow and a good second look through its writing would be useful. For example:

- RGB color is discussed, for example the color values are used to assign conductivities, yet I cannot figure out whether the algorithm just averages all three colour channels before starting, or will generate three maps from each channel. The maths all seem to pertain to greyscale images.
- Often things are described by analogy to slime molds. While this may be intuitive to the authors, for myself I have a lot more experience and appreciation of image processing techniques than I do of slime molds, and often find these analogies to be more distracting than helpful.
 - o e.g. in the first paragraph of II the authors discuss flow and consolidation onto a network-like structure; to me this sounds or feels like the watershed algorithm. I know it's not, that's made clear later, but that is the standard method that comes to mind when reading this. However, the description as a slime mold that starts by covering the whole domain doesn't mean much to me (is this how these species behave, generally? I thought they were networks?)
- terms like barycentre are confusing. Is this just the middle of each pixel?
- similarly, the second paragraph of II, with the pre-extracted network, is very confusingly written. I think its just thresholding the image, then connecting adjacent pixels that are above that threshold. It's not clear whether these are 8-neighbours or 4-neighbours, as written, although later figure suggests it's 8-neighbour connectivity.
- As a minor point, the conductivities are said to be the diameter of an edge, whereas if this is meant to mimic something like Darcy flow, then wouldn't it be the cross-sectional area (or a length squared)?
- ℓ_e is used in Equation 1, yet not defined till after Equation 4. Also, isn't this either going to be simply 1 or $\sqrt{2}$, depending on whether it's a neighbour or diagonal neighbour connection?
- When discussing the network transportation costs, a reference would be helpful, as this is certainly relying on a particular model with particular assumptions of cost.

Finally, the river data set is itself a bit odd. The figures are taken from a map (sea charts, including if you look at the online data, place names, borders, icons, and in one example a map 'pin'). While there is nothing wrong in principle with working with map data, it is going to be a bit artificial, as it has already basically been 'drawn' by hand. For something that would have better ecological/geophysical value, one would have expected satellite data (which is what I assumed the images were until I accessed them on github, then looked up the source). A better presentation of the data itself, at least, would be useful.

Decision letter (RSOS-210025.R0)

Dear Dr De Bacco

The Editors assigned to your paper RSOS-210025 "Principled network extraction from images" have now received comments from reviewers and would like you to revise the paper in accordance with the reviewer comments and any comments from the Editors. Please note this decision does not guarantee eventual acceptance.

Please submit your revised manuscript and required files (see below) no later than 21 days from today's (ie 28-May-2021) date. Note: the ScholarOne system will 'lock' if submission of the revision is attempted 21 or more days after the deadline. If you do not think you will be able to meet this deadline please contact the editorial office immediately.

on behalf of Marta Kwiatkowska (Subject Editor)
openscience@royalsociety.org

Associate Editor Comments to Author:

Comments to the Author:

Thank you for this submission. There are a number of points that the reviewers have identified as needing addressing. Please ensure you resolve these matters in a revision (and ensure that a full point-by-point response is included when you submit the revision) as - generally - we're only able to offer one round of revision.

Reviewer comments to Author:

Reviewer: 1

Comments to the Author(s)

Author(s) proposed Image2net model to extract networks from images. The topic is interesting but my comments on this article as follows:

1. add some more references related to network building from images.
2. Refine the introduction section in the standard format.
3. You may consider some statistical metrics for evaluating the performance of the image2net model.
4. How network gives the correct information? pls. give theoretical reason about getting the correct network. There are various deep learning technique like autoencoders for prediction the unlabeled images.
5. Need more discussion on results.
6. Also mention the limitation of the proposed model.

Reviewer: 2

Comments to the Author(s)

The article of Baptist and De Bacco describes a method of extracting a network from an image, where the network represents the connections between the key features in the image. This is a challenging problem and one with wide application. The methods presented are, in my experience, relatively innovative, and I would recommend publication in Open Science after addressing some concerns about the presentation of the method and results in the manuscript.

Most significantly, there are some quite strong assumptions in the way the image processing proceeds, which mean that it will be most appropriate to a particular class of phenomena with well-defined and physically meaningful flows or fluxes in it. Indeed, the three sample data sets include this (shuttle-flow in slime mold, blood flow in the retina, water flow in rivers; all nice conserved incompressible flows, with all the constraints that that implies). It would help potential users of this method to be explicit about this, for example in the abstract, intro and conclusions. While the methods here may be useful for other systems, the way the processing works really relies on mimicking the underlying transport, yet this is not stated clearly. Not all networks represent flows – fracture networks for example are well studied, or foams, or grain boundaries in materials. Even flow-based systems may also not be as well-defined as those studied here, for example tidal marshes, where the flow reverses several times a day (and which are more loopy than normal river/stream networks).

Another aspect of the processing that deserves comment, especially given some of the data sets used, is how it handles images with distinct (i.e. not topologically connected) objects in it. For example, an image containing parts of two different rivers. Unless I have missed something important, and the images provided in the SI don't seem to suggest that I have, the entire network found must belong to a single connected cluster of points. If this is indeed correct, it really needs to be discussed, as it will affect the applications that this code can be applied to.

For the method, the way it is presented is often unclear, or hard to follow and a good second look through its writing would be useful. For example:

- RGB color is discussed, for example the color values are used to assign conductivities, yet I cannot figure out whether the algorithm just averages all three colour channels before starting, or will generate three maps from each channel. The maths all seem to pertain to greyscale images.
- Often things are described by analogy to slime molds. While this may be intuitive to the authors, for myself I have a lot more experience and appreciation of image processing techniques than I do of slime molds, and often find these analogies to be more distracting than helpful.
 - o e.g. in the first paragraph of II the authors discuss flow and consolidation onto a network-like structure; to me this sounds or feels like the watershed algorithm. I know it's not, that's made clear later, but that is the standard method that comes to mind when reading this. However, the description as a slime mold that starts by covering the whole domain doesn't mean much to me (is this how these species behave, generally? I thought they were networks?)
- terms like barycentre are confusing. Is this just the middle of each pixel?

- similarly, the second paragraph of II, with the pre-extracted network, is very confusingly written. I think its just thresholding the image, then connecting adjacent pixels that are above that threshold. It's not clear whether these are 8-neighbours or 4-neighbours, as written, although later figure suggests it's 8-neighbour connectivity.
- As a minor point, the conductivities are said to be the diameter of an edge, whereas if this is meant to mimic something like Darcy flow, then wouldn't it be the cross-sectional area (or a length squared)?
- ℓ_e is used in Equation 1, yet not defined till after Equation 4. Also, isn't this either going to be simply 1 or $\sqrt{2}$, depending on whether it's a neighbour or diagonal neighbour connection?
- When discussing the network transportation costs, a reference would be helpful, as this is certainly relying on a particular model with particular assumptions of cost.

Finally, the river data set is itself a bit odd. The figures are taken from a map (sea charts, including if you look at the online data, place names, borders, icons, and in one example a map 'pin'). While there is nothing wrong in principle with working with map data, it is going to be a bit artificial, as it has already basically been 'drawn' by hand. For something that would have better ecological/geophysical value, one would have expected satellite data (which is what I assumed the images were until I accessed them on github, then looked up the source). A better presentation of the data itself, at least, would be useful.

===PREPARING YOUR MANUSCRIPT===

===PREPARING YOUR REVISION IN SCHOLARONE===

Author's Response to Decision Letter for (RSOS-210025.R0)

See Appendix A.

RSOS-210025.R1 (Revision)

Review form: Reviewer 1 (Sushil Kumar)

Is the manuscript scientifically sound in its present form?

Yes

Are the interpretations and conclusions justified by the results?

Yes

Is the language acceptable?

Yes

Do you have any ethical concerns with this paper?

No

Have you any concerns about statistical analyses in this paper?

Yes

Recommendation?

Accept as is

Comments to the Author(s)

Dear authors,

I appreciate your work in the field of image processing in which you have presented a new model that builds the networks from images.

Review form: Reviewer 2

Is the manuscript scientifically sound in its present form?

Yes

Are the interpretations and conclusions justified by the results?

Yes

Is the language acceptable?

Yes

Do you have any ethical concerns with this paper?

No

Have you any concerns about statistical analyses in this paper?

No

Recommendation?

Accept as is

Comments to the Author(s)

The authors have addressed all my concerns. The results are an interesting and innovative contribution to the image processing literature, and with open code and data represent the best traditions of open science.

Decision letter (RSOS-210025.R1)

Dear Dr De Bacco,

It is a pleasure to accept your manuscript entitled "Principled network extraction from images" in its current form for publication in Royal Society Open Science. The comments of the reviewer(s) who reviewed your manuscript are included at the foot of this letter.

Kind regards,

Royal Society Open Science Editorial Office
Royal Society Open Science
openscience@royalsociety.org

on behalf of Professor Marta Kwiatkowska (Subject Editor)
openscience@royalsociety.org

Reviewer comments to Author:

Reviewer: 1
Comments to the Author(s)

Dear authors,

I appreciate your work in the field of image processing in which you have presented a new model that builds the networks from images.

Reviewer: 2
Comments to the Author(s)

The authors have addressed all my concerns. The results are an interesting and innovative contribution to the image processing literature, and with open code and data represent the best traditions of open science.

Appendix A

Response to the reviewers

We thank the reviewers and editor for their positive and constructive feedback. Your comments and suggestions have significantly improved our paper.

We would like to highlight the following changes in this revision:

- Added discussion about limits of our model.
- Clarified various definitions of quantities and algorithmic steps as raised by the reviewers.
- Revised the analogy with the slimed mold to keep it self-contained in the introduction.
- Added more references to techniques for networks extraction from images.

We give our detailed response below.

Reviewer 1

Author(s) proposed Image2net model to extract networks from images. The topic is interesting but my comments on this article as follows:

1. add some more references related to network building from images.

We added the references Hu et al. (2007); Tupin et al. (1998); Bastani et al. (2018); Wang et al. (2016); Montoya-Zegarra et al. (2019); Rapacz and Łazarz (2020); Price (2012); Fraz et al. (2012) which use image processing techniques for road, leaf venation, cardiovascular and retinal network extraction.

2. Refine the introduction section in the standard format.

We are not sure about what the reviewer exactly means with this comment. Our introduction is currently structured with: definition of the problem, challenges, description of what we do; related work; attach to this the main approach from the literature where our model is based and show the specificity of adapting this problem on images. We believe that this is a standard format, but are open to make changes in case the reviewer could give us specific indications for this.

3. You may consider some statistical metrics for evaluating the performance of the image2net model.

We evaluate the performance of our algorithm by i) computing the similarity measure (Equation (8), presented in Baptista et al. (2020)) between the extracted and the ground-truth networks for a set of 40 images (provided by Hoover et al. (2000)), and by ii) comparing its accuracy to that of the other methods in a set of 55 images associated to different structures (*rivers, retina, Physarum Polycephalum*). These have been both validated and utilized in the works cited, we believe that they are valid statistical metrics for performance evaluation.

4. How network gives the correct information? pls. give theoretical reason about getting the correct network. There are various deep learning technique like autoencoders for prediction the unlabeled images.

We believe that defining an extracted network as 'correct' is a strong statement, in the absence of ground truth. In fact, this is an unsupervised problem where one has two possible ways to validate the performance. If one has a baseline for comparison, e.g. in the retina dataset where we had access to hand-labeled networks, this can be used as candidate 'ground-truth'. Notice however that also in this case there could be human bias or errors in the baseline for comparison. Alternatively, in the absence of 'ground-truth' candidates, one can compare the values on pixels with the edge weights extracted, which we do with the similarity measure. We are not sure how one could use deep learning techniques like auto-encoders to solve this problem, as

they require a set of labeled examples (that we may not have) and would require big training sets to estimate the parameters, which maybe be not available for the types of datasets considered here.

■ *5. Need more discussion on results.*

We discussed results in three different ways: i) in the presence of ground-truth candidates; ii) in the absence of it; and iii) qualitatively. We believe that these together give a comprehensive picture.

■ *6. Also mention the limitation of the proposed model.*

We added a discussion about this by addressing a comment from referee 2. Specifically, we discussed about the validity of our model for incompressible flows and in the presence of many loopy structures.

Reviewer 2

The article of Baptista and De Bacco describes a method of extracting a network from an image, where the network represents the connections between the key features in the image. This is a challenging problem and one with wide application. The methods presented are, in my experience, relatively innovative, and I would recommend publication in Open Science after addressing some concerns about the presentation of the method and results in the manuscript.

We thank the reviewer for appreciating the innovative feature of our work, and for recommending it for publication in Open Science. We addressed their concerns which have improved our paper.

Most significantly, there are some quite strong assumptions in the way the image processing proceeds, which mean that it will be most appropriate to a particular class of phenomena with well-defined and physically meaningful flows or fluxes in it. Indeed, the three sample data sets include this (shuttle-flow in slime mold, blood flow in the retina, water flow in rivers; all nice conserved incompressible flows, with all the constraints that that implies). It would help potential users of this method to be explicit about this, for example in the abstract, intro and conclusions. While the methods here may be useful for other systems, the way the processing works really relies on mimicking the underlying transport, yet this is not stated clearly. Not all networks represent flows — fracture networks for example are well studied, or foams, or grain boundaries in materials. Even flow-based systems may also not be as well-defined as those studied here, for example tidal marshes, where the flow reverses several times a day (and which are more loopy than normal river/stream networks).

Thanks for this excellent comment. We added an extensive discussion about this in the conclusion, pointing out at the limitations following the referee's insights. In particular, we stated that while our method is still applicable in all the scenarios pointed out by the referee, the extracted networks might not be meaningful. We thus encourage the practitioners to select the input images consciously, keeping this in mind.

Another aspect of the processing that deserves comment, especially given some of the data sets used, is how it handles images with distinct (i.e. not topologically connected) objects in it. For example, an image containing parts of two different rivers. Unless I have missed something important, and the images provided in the SI don't seem to suggest that I have, the entire network found must belong to a single connected cluster of points. If this is indeed correct, it really needs to be discussed, as it will affect the applications that this code can be applied to.

Thanks for pointing this out, this deserves indeed a clarification. Our algorithm

can deal with images displaying different objects resulting in different connected components in two ways: i) by properly placing sources and sinks in the different connected components of the network shown in the image. In fact, depending on how sources and sinks are selected, our algorithm can give in output naturally more than one connected component. Alternatively, ii) one can apply the algorithm separately in each of the different connected components. We have added a brief discussion about this in the main draft.

For the method, the way it is presented is often unclear, or hard to follow and a good second look through its writing would be useful. For example: RGB color is discussed, for example the color values are used to assign conductivities, yet I cannot figure out whether the algorithm just averages all three colour channels before starting, or will generate three maps from each channel. The maths all seem to pertain to greyscale images.

We added the exact mapping from the RGB decomposition into an integer value to the Supplementary Information.

Often things are described by analogy to slime molds. While this may be intuitive to the authors, for myself I have a lot more experience and appreciation of image processing techniques than I do of slime molds, and often find these analogies to be more distracting than helpful. o e.g. in the first paragraph of II the authors discuss flow and consolidation onto a network-like structure; to me this sounds or feels like the watershed algorithm. I know it's not, that's made clear later, but that is the standard method that comes to mind when reading this. However, the description as a slime mold that starts by covering the whole domain doesn't mean much to me (is this how these species behave, generally? I thought they were networks?)

Thanks for pointing this out, we understand that the analogy with the slime mold may sound distracting, hence we only kept it in one paragraph inside the introduction (to introduce intuitively the main quantities, conductivities and flows) but then removed references to it in the remaining of the text.

terms like barycentre are confusing. Is this just the middle of each pixel?

We clarified this in the paper by mentioning center of mass instead of barycenter. Hopefully, this should be more clear.

similarly, the second paragraph of II, with the pre-extracted network, is very confusingly written. I think its just thresholding the image, then connecting adjacent pixels that are above that threshold. It's not clear whether these are 8-neighbours or 4-neighbours, as written, although later figure suggests it's

■ *8-neighbour connectivity.*

Thanks for pointing this out, we clarified this in the paper by specifying how edges are built and adding a reference to the terminology used in the paper where this routine is defined in detail.

■ *As a minor point, the conductivities are said to be the diameter of an edge, whereas if this is meant to mimic something like Darcy flow, then wouldn't it be the cross-sectional area (or a length squared)?*

Your interpretation is correct. We changed to “The conductivities can be interpreted as *proportional to* the size of the diameter of an edge”, as the exact relationship may depend on the application.

■ *l_e is used in Equation 1, yet not defined till after Equation 4. Also, isn't this either going to be simply 1 or $\sqrt{2}$, depending on whether it's a neighbour or diagonal neighbour connection?*

Thanks for pointing this out. This is correct, in this particular case this is 1 or $\sqrt{2}$, but we prefer to keep l_e in the equations to consider more general cases where the topology may not be as regular; for instance, when nodes are not the center of mass of pixels. We added this clarification in the paper and defined it after Eq (1).

■ *When discussing the network transportation costs, a reference would be helpful, as this is certainly relying on a particular model with particular assumptions of cost.*

We added a reference to Baptista et al. (2020) that contains a detailed explanation of this cost.

■ *Finally, the river data set is itself a bit odd. The figures are taken from a map (sea charts, including if you look at the online data, place names, borders, icons, and in one example a map 'pin'). While there is nothing wrong in principle with working with map data, it is going to be a bit artificial, as it has already basically been 'drawn' by hand. For something that would have better ecological/geophysical value, one would have expected satellite data (which is what I assumed the images were until I accessed them on github, then looked up the source). A better presentation of the data itself, at least, would be useful.*

Thank you for pointing this out. We added a section to the Supplementary Information called River satellite images, where we show a network extracted using a satellite river image as input and show how results are similar as those obtained for the river dataset.

References

- Baptista, D., Leite, D., Facca, E., Putti, M., and De Bacco, C. (2020). Network extraction by routing optimization. *Scientific Reports volume*, 10(20806).
- Bastani, F., He, S., Abbar, S., Alizadeh, M., Balakrishnan, H., Chawla, S., Madden, S., and DeWitt, D. (2018). Roadtracer: Automatic extraction of road networks from aerial images. In *Proceedings of the IEEE Conference on Computer Vision and Pattern Recognition*, pages 4720–4728.
- Fraz, M. M., Remagnino, P., Hoppe, A., Uyyanonvara, B., Rudnicka, A. R., Owen, C. G., and Barman, S. A. (2012). Blood vessel segmentation methodologies in retinal images—a survey. *Computer methods and programs in biomedicine*, 108(1):407–433.
- Hoover, A., Kouznetsova, V., and Goldbaum, M. (2000). Locating blood vessels in retinal images by piecewise threshold probing of a matched filter response. *IEEE Transactions on Medical Imaging*, 19:203–210.
- Hu, J., Razdan, A., Femiani, J. C., Cui, M., and Wonka, P. (2007). Road network extraction and intersection detection from aerial images by tracking road footprints. *IEEE Transactions on Geoscience and Remote Sensing*, 45(12):4144–4157.
- Montoya-Zegarra, J. A., Russo, E., Runge, P., Jadhav, M., Willrodt, A.-H., Stoma, S., Nørrelykke, S. F., Detmar, M., and Halin, C. (2019). Autotube: a novel software for the automated morphometric analysis of vascular networks in tissues. *Angiogenesis*, 22(2):223–236.
- Price, C. A. (2012). Leaf gui: analyzing the geometry of veins and areoles using image segmentation algorithms. In *High-throughput phenotyping in plants*, pages 41–49. Springer.
- Rapacz, M. and Łazarz, R. (2020). Automatic extraction of leaf venation complex networks. In *ECAI 2020*, pages 1914–1921. IOS Press.
- Tupin, F., Maitre, H., Mangin, J.-F., Nicolas, J.-M., and Pechersky, E. (1998). Detection of linear features in sar images: Application to road network extraction. *IEEE transactions on geoscience and remote sensing*, 36(2):434–453.
- Wang, W., Yang, N., Zhang, Y., Wang, F., Cao, T., and Eklund, P. (2016). A review of road extraction from remote sensing images. *Journal of traffic and transportation engineering (english edition)*, 3(3):271–282.